# Regulatory Compliance in Online Dog Advertisements in Australia

**DOI:** 10.3390/ani10030425

**Published:** 2020-03-03

**Authors:** Ana Goncalves Costa, Torben Nielsen, Eleonora Dal Grande, Jonathan Tuke, Susan Hazel

**Affiliations:** 1School of Animal & Veterinary Sciences, University of Adelaide, 5005 Adelaide, Australia; torben.nielsen@adelaide.edu.au (T.N.); eleonora.dalgrande01@adelaide.edu.au (E.D.G.); susan.hazel@adelaide.edu.au (S.H.); 2School of Mathematical Sciences, University of Adelaide, 5005 Adelaide, Australia; simon.tuke@adelaide.edu.au

**Keywords:** dog, online, Australia, regulation, microchipping, desexing, breeder ID, advertising

## Abstract

**Simple Summary:**

As Australians increasingly purchase their companion dogs online, Australian state and territory authorities are faced with the challenge of ensuring online sales adhere to local regulations. Using webscraping techniques, we analysed 1735 unique advertisements for dogs and puppies from Gumtree—one of Australia’s most popular trading platforms—and benchmarked levels of microchipping, desexing and breeder identification numbers in each state and territory. We found an increased likelihood of microchipping in states requiring microchipping prior to sale and inclusion of chip numbers in advertisements. Older animals who were microchipped were more likely to be desexed, and advertisements placed by breeders who were selling vaccinated puppies were more likely to include a breeder registration number in their ad than sellers who identified themselves as owners. We recommend regulatory bodies use this data to make evidence-based decisions on future regulation and use this benchmark to monitor effectiveness of any changes.

**Abstract:**

In Australia, each state and territory authority implements and enforces regulations regarding dog management—including the breeding and sale of dogs online—which is increasingly becoming the most popular method of obtaining pets. The aims for this study included: 1. Benchmarking regulatory compliance in online dog advertisements in Australia, and, 2. Understanding factors associated with regulatory compliance in online advertisements. We collected advertisements for dogs and puppies from Gumtree—one of Australia’s most popular online trading platforms—on two separate days, two weeks apart (25 March and 8 April 2019). A total of 1735 unique advertisements were included in the dataset. Chi-squared tests and multivariable logistic regression models were used to identify risk factors for microchipping, desexing and breeder identification number, and compliance levels. State laws requiring animals to be microchipped prior to sale and the inclusion of chip numbers in advertisements were found to be the biggest factor in increasing likelihood of microchipped animals in Gumtree advertisements, while desexing was more common in microchipped and older animals. The online ad was more likely to include a breeder ID if the dog was young, vaccinated, and advertised by a breeder rather than an owner. The findings from this study will assist regulators to make evidence-based decisions on managing online advertisements for companion animals. In the future, the benchmarking this study has presented will allow future analysis of the effectiveness of regulation changes.

## 1. Introduction

More than half of all Australian homes are shared with a companion animal, almost 40% of which do so with at least one dog [1], compared to 27% of the UK and 50% of USA households, respectively [2]. With the total population of dogs reaching 5 million nationwide, and increasing by 3% each year, Australia’s love for dogs feeds a 13 billion-dollar industry [1] and requires a supply of approximately 480,000 animals each year to meet demand [3].

Purchasing dogs online is becoming increasingly popular worldwide [4] with 57% of European buyers now using the internet to source their companion animals [5]. There is currently inadequate research on Australian online markets, and no literature on the percentage of Australians who choose online platforms to search for, and purchase, their companion animals. However, 73% of Australians shopped online in 2018 [6] and it is predicted that one in 10 items will be bought online by 2020 [7]. With the growing popularity of online purchases in Australia, regulations are required to manage the changing marketplace. Online transactions are inherently less transparent, and harder to track than traditional physical trading structures, i.e., physical stores, creating unique challenges for regulators [8,9].

In 2001, the Pet Advertising Advisory Group (PAAG) was founded to fill a gap in the lack of online animal advertising in the UK. It engages with online marketplaces and asks that they commit to only hosting advertisements which follow PAAG’s minimum standards, including removing any advertisements selling un-microchipped animals [10]. There is currently no equivalent entity or standards in either Australia, Europe, nor the USA, all of which lack a country-wide unified plan for regulating online pet sales. While PAAG members enforce their standards voluntarily, and therefore rely on the willingness of each individual website to comply, it does provide a starting point in educating websites, sellers and buyers in best practice regarding animal trade. Future research into the differences between PAAG compliant sites and non-compliant sites would likely prove indispensable.

In Australia, each state and territory authority decides on and enforces regulations relating to companion animal management, therefore each state and territory differ in their requirements resulting in considerable variability of laws within the country (see Table 1). This variability is heightened due to a number of factors including which department is designated with the implementation of these regulations (i.e., agricultural or environmental departments), whether there is strong community support, and whether local political parties have the will to regulate it.

In recent years many state and territory authorities have updated their regulations surrounding animal management, most notably, in July 2019 (several months after data for this study was collected), Victoria launched their ‘Pet Exchange Register’ and overhauled their animal management regulations, adding the requirements of including ‘source numbers’, alongside microchip numbers for animals in all advertisements, and allowing members of the public to report non-compliant advertisements. This online register also allows buyers to confirm the validity of source numbers prior to making purchase decisions [11].

Regulation in the animal trade industry centers on the protection of animals and buyers in the market as well as managing Australia’s companion animal population. Microchipping and breeder identification numbers serve to identify and monitor breeding facilities and track animals back to their original breeders. Desexing is designed to eliminate the possibility of further breeding by individuals outside the industry and managing the total dog population in Australia. Regulation compliance, then, is required to effectively manage animal trade.

Microchipping a dog is an example of ‘good guardianship’ behaviour. Rohlf, et al. [12], have shown microchipping is influenced mostly by the owner’s normative beliefs, and non-compliance may be due to the perceived difficulty of microchipping by owners, leading to a recommendation that regulations should require microchipping prior to purchase. The median age of microchipping in Australia is 74.4 days for dogs (just under 11 weeks), and dogs are often microchipped at time of vaccinations [13]. Animal Medicine’s Australia’s (AMA) 2016 report [2] showed that Victoria had the highest rate of microchipping in Australia, at 89%.

Owners are more likely to desex their dog if there is a general positive attitude towards desexing practices within the individuals’ societal group, and similar to microchipping, normative beliefs play a crucial role in the choice to desex [12,14]. This is despite a quarter (25.7%) of Australians surveyed stating that “desexing is unnatural” and more than half (55.5%) believing it should not be regulated. Rohlf, et al., [12] also found that older dogs were more likely to be desexed than younger dogs, and pure breed dogs were less likely to be desexed. They also reported that the belief desexing would be beneficial for behavioural reasons increased the likelihood of owners opting in. Although early-age desexing does occur, and is advocated by the Royal Society for the Prevention of Cruelty to Animals (RSPCA) [15], the risks and benefits of desexing, particularly in young animals, is still contested [16], and requiring desexing prior to sale is unlikely to be an option for future regulation for puppies.

Only three states require breeders to have identification numbers (ID) and each state differs in their requirements. The State of South Australia (SA) requires all breeders (whether professional, or accidental breeding) to register their details online to receive an identification number which they then must place on all advertisements. Alternatively, SA breeders may use their Australian National Kennel Council (ANKC) member number as their breeder ID number. New owners, if they choose to sell their dog in the future, are required to include the original breeder name and ID number in all new advertisements. This applies for all dog born after 1 July 2018. The State of Queensland (QLD) requires an ID number for all dogs born on or after 26 May 2017, with exceptions for working dogs. Six types of ID’s are accepted by QLD as a valid supply number, including, but not limited to; breeder ID numbers; ANKC registration numbers, and; Queensland Racing Integrity Commission numbers. The Victorian State (VIC) only requires registration for breeders housing more than three fertile dogs and does not require the breeder ID number to be placed in advertisements.

Increasing traceability of animals in the animal companion market is particularly important when puppy factories (also known as “puppy farms”, or “puppy mills”) are considered, which has become a concern with online transactions [9,17]. Puppy factories are defined by their priority of profit, at the expense of the animals’ welfare, health, and temperament [18]. The health and behaviour consequences of poor breeding practices is well documented in literature and can include; higher incidences of behaviour issues [19], including fear, phobias [20], aggression [21], and; significantly poorer health [20,21,22], increased chronic pain and skin issues [23], ultimately leading to; shorter lifespans [23]. Regulations on dog and puppy trade has been declared both nationally, and internationally, as the most effective way to combat puppy factories [5,24] with support from RSPCA Australia [24,25] and the Australian community [26,27].

Gumtree is an Australian online marketplace providing free and paid advertising options for sellers for a variety of different listings, including dogs and cats. Seven million Australians visit the platform each month and 85,000 advertisements are posted each day [28]; it is one of the largest trading websites in Australia. There is limited information regarding animal online trading regulation compliance worldwide, but Australian studies are particularly lacking. The aims for this study included; 1. Benchmarking regulatory compliance in online dog advertisements in Australia, and; 2. Understanding factors associated with regulatory compliance in online advertisements.

## 2. Materials and Methods

### 2.1. Extraction of Gumtree Advertisement Data

Two webscrapes were performed two weeks apart, 25 March and 8 April 2019 using the rvest package [29] in R [30]. Variables included: type of animal for sale, title of ad, location of animal (map and state/territory), owner status (owner/breeder/shelter/rescue), ad text, price, negotiable and urgent status, date ad was listed, animal date of birth and microchip, vaccination, vet history, and desexing checkbox status. This study was approved by the Human Ethics Committee of the University of Adelaide (H-2019-043).

### 2.2. Management of Data

Webscraping yielded 907 advertisements on 25 March 2019, and 928 advertisements on 8 April 2019. Microsoft Excel^®^ was used for data management, removing advertisements selling services, products, or animals not identified as dogs. The number of animals available in each ad were counted using the ad text. Animals clearly labelled as sold or on hold were excluded. If insufficient data was available, number was labelled as unspecified. Duplicates were identified as two or more advertisements with identical date of birth, breed and location, and these were deleted. In each case the oldest ad was removed.

For advertisements which did not clearly identify a state or territory, the map data variable, in conjunction with suburbs and postcodes [31], was used to confirm listed town location. Additionally, ANKC registration details were used to identify, or confirm, the state from which the ad was placed, e.g., Dog and Cats Online registration numbers (DACO) were SA, DogsWest affiliates were counted as Western Australia (WA). If insufficient data was available to identify the State or Territory, the ad was classified as ‘unknown’.

Breeds were identified either through the ad title, or ad text. Purebred was assumed where only one breed was identified in the ad text and title. Advertisements which identified more than one breed for a single dog were listed as cross breed, using the first breed specified as the cross. Any breed specifically given a marketing name—i.e., Frug, or Groodle—or which specifically identified two purebred parents of different breeds, were labelled as a designer breed. For advertisements with more than one dog, the data of the first animal described was used. If no breed was specified for puppies, the dams breed was used. If insufficient breed data was provided, breed was classified as unknown. Coat colour and length were not deemed differential breed identifiers, however, size (i.e., toy, mini) was.

Microchipping, vaccination, desexing, and veterinary checks were confirmed if the seller checked any of the relevant ad description boxes or specifically stated so in the ad text. Descriptions either specifically stating ANKC membership or providing ANKC member numbers were deemed current ANKC members. DACO registration number and breeder ID numbers alongside council registration numbers (i.e., for breeders with more than three fertile dogs in Vic) were considered state registered.

Prices were kept as in the original ad, excluding those obviously artificial, e.g., $1234 (4 records), and those priced at $1 (11 records), which were labelled as missing, leading to total of 52 advertisements with missing price details. Dates of birth were changed only in the case of outliers that were clearly incorrect when compared to ad descriptions and left as unspecified if not enough data was available to rectify. Advertisements with animal ages less than 0 days were removed.

Breeder organisation affiliation included any ad which named a breeding organisation in the ad text, including but not limited to, ANKC and its state affiliates, i.e., DogsSA.

The regulations for this study was based on regulations as of 25 March 2019, see Table 1.

### 2.3. Statistical Analysis

Analysis of data was completed in R, version 3.6.0—“Planting of a Tree” (The R Foundation, Vienna, Austria). Frequency analysis was completed prior to full statistical analysis and chi-squared tests or Fisher’s exact tests were performed on categorical data to test for association between variables. In the first model, microchip status was the dependent variable while desexing status and breeder ID number in the ad text were independent variables. For the second model, desexing status was the dependent variable, while microchip and breeder ID number in the ad text were independent variables. The final model used inclusion of breeder ID number as the dependent variable and microchip and desexing status as independent variables. A *p*-value of <0.05 was considered significant in all statistical analysis.

Multivariable logistic regression modelling was used to identify risk factors for microchipping and desexing using backwards stepwise regression methods with nonsignificant variables removed from the model. Variables with chi-squared test *p*-values less than 0.25 were included in the regression models, and subsequently removed based on the log likelihood ratio test. Based on literature and plausibility, the interaction between desexing status and microchipping, due to the age differences of when these procedures typically occur, lead to the decision to not include desexing in the microchipping model. To assess the fit of the multivariable regression models we used Akaike Information Criterion (AIC) statistics. All models excluded the territories, Australian Capital Territory (ACT) and Northern Territory (NT), due to counts being too small for meaningful analysis. Dog breed was not included due to the high numbers of different breeds. Breeder ID number models only included the states with relevant systems to issue ID’s to breeders. As counts were too small for analysis, ‘unknown’ was changed to ‘no’ for microchipping, vaccination and breeder org affiliation.

## 3. Results

### 3.1. Descriptive Statistics

A total of 1735 valid, cleaned advertisements comprised the dataset for the analysis in this study. These advertisements represented the sale of 3836 dogs and puppies, with 840 (48.4%) advertisements offering an individual dog and, 718 (41.4%) of advertisements offering more than one. A total of 177 (10.2%) sellers advertised more than one dog without specifying numbers.

The most common breeds available for sale were American Staffordshire Terrier (7.5%), French Bulldog (7.3%), Staffordshire Terrier (6.7%), Kelpie (5.3%) and the German Shepherd (5.07%). While American Staffordshire terrier was most common dog Australia-wide, French bulldogs were the most common in the State of New South Wales (NSW), Queensland and Victoria. Within the dogs over 6 months of age (N = 481) the most common breeds were Staffordshire terrier (7.5%), American Staffordshire terrier (7.3%), and German Shepherd (6.2%). For dogs under 6 months of age, the most common breeds were the French Bulldog (8.8%), American Staffordshire terrier (7.7%), and Staffordshire terrier (6.5%). See Appendix A for complete list of breeds.

NSW had the highest proportion of advertisements (33.1%), followed by QLD (28.6%), see Table 2. VIC had the highest rate of microchipping, and all states had similar rates of desexing. 

Of the total advertisements, 89.9% listed a sale price, creating a total value of $4,662,080. Purebred dogs had higher median prices than cross breeds, while designer dogs had the highest median prices (see Table 3). See Appendix A for top 20 breed cross, pure and designer percentages. Younger dogs were more likely to be sold at higher prices than older dogs. There were 99 advertisements (5.7%) with dogs listed as free, 52.5% of free advertisements were for dogs between 1 and 7 years of age. A high proportion of designer dogs were under 6 months of age (90.4%) versus cross breeds (65.4%), and purebreds (68.8), see Table 3.

More than one in 10 advertisements (11.2%) indicated association to ANKC or its local affiliate groups, while 17% indicated association to other breeder organisations, i.e., Australian Association of Pet Dog Breeders, Master Dog Breeders and Associates, and Responsible Pet Breeders Australia. The majority of advertisements (71.5%) made no mention of breeder association membership. Of the 490 advertisements affiliated with breeder organisations, 386 (78.8%) included a breeder membership ID number in the ad text.

Univariate analysis found that advertisers who identified themselves as breeders were 61% more likely to sell microchipped dogs than those who identified as owners, but owners were 87% more likely to advertise desexed animals compared to those identifying as breeders; see Table 4 and Table 5. Those who identified themselves as breeders were 71% more likely to include a breeder ID in their advertisement than other types of sellers, see Table 6.

Victoria had the highest rate of microchipped dogs for sale (92.7%) and the highest rate of microchip numbers included in advertisements (69.5% compared to a range between 0.0% and 6.1% for other state and territories) but held the lowest rate of advertisements which including a breeder ID number in Australia (5.1%), see Table 7.

#### 3.1.1. Microchipping

Shelter and rescue advertisements were excluded from the microchipping multivariable logistic regression model due to high levels of correlation (standard rehoming policies dictate all rescue and shelter animals are desexed and microchipped prior to sale [1]). Univariate analyses found that desexed dogs were 15% more likely to be microchipped. However, while desexing was found to be a significant factor in the regression model, desexing was subsequently removed from the model due to being highly correlated with the age of the dog.

Dogs advertised in advertisements posted in VIC were 67% more likely than NSW and over 80% more likely than SA and Tasmania (TAS) to be microchipped (see Figure 1). An owner was 26% less likely to sell a microchipped dog on Gumtree than a breeder, and the more expensive the dog, the more likely they were to be microchipped.

#### 3.1.2. Desexing

Desexing is regulated in two state/territories, SA and ACT. Neither jurisdiction had enough dog advertisements fall under their regulations to enable analysis between regulated state/territories and unregulated state/territories.

Only dogs over the age of six months were used in the multivariable logistic regression model. Shelter and rescue advertisements were excluded from the desexing multivariable logistic regression model due to interactions. ACT and NT were also excluded due to the low advertisement count, leaving a total of 499 advertisements and of these advertisements, 22% advertised desexed dogs. Seller, vaccination and breeder organisation affiliation were found to be nonsignificant. The biggest predictor of a dog being desexed prior to sale was microchipping, which was associated with a 168% increased likelihood of desexing (see Figure 2). Advertisements selling older dogs were also more likely to indicate a desexed animal than younger ones and dogs with a sale price of over $2001 were less likely to be desexed.

#### 3.1.3. Breeder ID

Only advertisements from states regulating breeder registration (breeder identification number) were included in the breeder ID multivariable logistic regression model (SA, QLD, VIC). ACT also required breeder ID’s, however, there were too few advertisements for analysis to be meaningful and the territory is therefore excluded. Two advertisements were unclear on whether they had a breeder ID number and were therefore excluded from the dataset for this model. Shelter and rescue advertisements are were excluded from this analysis. For SA, only dogs born on or after 1 July 2018 were included, and for QLD, only dogs born on or after 26 May 2017 were included, as regulations do not apply to animals born prior. Dogs for all states with unclear ages were removed due to low counts.

Of the 648 advertisements in this analysis, VIC had the lowest percentage of breeder ID numbers included, with 5.1% of sellers including a breeder ID number (see Figure 3).

Pure, cross, and designer breeds were found to be nonsignificant, as was microchipping and breeder organisation affiliation. The biggest predictor of inclusion of a breeder ID number in an advertisement was the age of the dog, with advertisements featuring dogs less than 8 weeks of age 47% more likely to include a breeder ID than dogs between 6 months and a year, and 92% more likely than advertisements featuring dogs over 3 years of age. Advertisements with vaccinated dogs were also 27% more likely to include a breeder ID number in their advertisements. Price was found to be significant in the model, however, the 95% CI for each price category did include one.

## 4. Discussion

### 4.1. Gumtree Advertisements

This study has benchmarked regulation compliance in Australian online (Gumtree) advertising, and explored the factors related to regulatory compliance.

NSW is Australia’s most populated state with over 8 million residents [32], it is therefore not unexpected that it featured as the state with the most Gumtree advertisements. Both VIC and NSW have over 9000 ANKC members, while QLD has only 6205 members [33]. Hazel et al. [34] also found NSW (35.6%) and QLD (34.8) to be the top two states for Gumtree advertisements for relinquished dogs and cats.

American Staffordshire terriers and Staffordshire bull terriers were the most common breeds in the Gumtree advertisements analysed. This is unsurprising, as they are considered some of the most popular breeds in Australia [1]. Some of the dogs labelled as ‘Staffy’ type dogs were cross breeds (see Appendix A). It is possible that some of these dogs are not in fact ‘Staffy’ breeds, as even experienced people are inaccurate when labelling ‘Staffy’ type breeds by appearance alone [35].

In this study, designer breeds made up almost one in 10 advertisements (7.3%) which appears to conflict with the findings of Hazel et al., [34], who did not report any designer dogs in their data, suggesting current data may be overrepresenting designer breeds in the market. However, the population represented in Hazel et al., [34] only included relinquished dogs andhe AMA 2019 report found a major increase in designer breed popularity in Australia in the last 3 years, from 8% to 14% [1]. It was unexpected to find designer dogs had higher sale values than purebreds. Beverland et al. [36] reported that dog ownership can be extrinsically motivated, in particular via a desire for status and/or self-esteem and found designer breeds were more likely to be chosen by these owners, which may explain the high price tag.

The 2019 AMA report showed approximately three in ten dogs in Australia were obtained for free [1]. Only 5.7% of the advertisements in this study were free, suggesting most free pets are traded by other methods, such as directly person to person.

Overall, 34.5% of sellers advertised puppies less than 8 weeks of age, although it was outside the scope of this study to quantify this data, most advertisements for underage puppies specified they would be “available in x weeks”. However, not all had this specification and it is likely that some would have been available prior to reaching 8 weeks of age. It is well documented that early separation from mum risks long-term behavioural concerns [37] and further research on how many underage puppies are available on Gumtree, and other online platforms, is recommended.

### 4.2. Microchipping

The likelihood of microchipping prior to sale on Gumtree increased if the advertisement was placed by a seller in Victoria; the dog or puppy was vaccinated; the seller identified themselves as a breeder, and when; the dog or puppy was sold at a higher price range.

VIC had the most stringent microchipping laws, requiring all sellers to microchip prior to sale and include microchip numbers for each individual dog available for sale. WA, TAS, and NT are the only states and territory that do not require microchipping prior to sale (3 months, 6 months, and no requirement, respectively), and have some of the lowest total microchipping, and compliance rates in Australia.

This study has revealed that dogs advertised as vaccinated are more likely to be microchipped, suggesting that sellers who take their animals to the vet are more likely to carry out both these procedures, supporting survey findings by McGreevy et al. [13]. The thoughts of Rohlf et al. [12] on normative beliefs are also supported by this study’s findings on Gumtree advertisements; sellers who comply, or pursue “good guardianship” practices, are more likely to comply fully, while those that are either unwilling, or unable to, do not comply at all. Whether non-compliance in Gumtree advertisements is due to financial, unwillingness, or access concerns is unknown and more research would be required before reaching any conclusions.

The results in the current study reflect AMA’s 2016 report where they reported Victoria had the highest rate of microchipping at 89% [2], similar to this study’s finding of 92.7% for online advertisements, indicating that Gumtree compliance in VIC is as high, if not, slightly higher than the state average. Australia’s overall microchipping rates have remained steady over the last few years at 86% [1], while only 70.3% of total Gumtree advertisements advertise microchipped animals. This discrepancy might be explained due to the differing regulations between states leading to a difference in microchip rates overall, i.e., WA and TAS do not require desexing until the dog is 3 and 6 months, respectively, leading to a large number of dogs under 3 and 6 months (which makes up the majority of advertisements) un-microchipped.

In 2013, Lancaster et al. [38] found that only 28% of stray dogs were microchipped when they entered the RSPCA Queensland shelter. This may show that stray dogs are more likely to come from households who are less likely to microchip their dogs, either due to lack of knowledge, financial ability or willingness to do so. It must also be noted that some dogs who are microchipped may bypass the RSPCA, as their microchips can be scanned at local vets (where stray animals are often taken to by members of the public), leaving animals who are more difficult to identify (un-microchipped) more likely to enter the RSPCA system, potentially showing a skewed population. However, correct and updated microchip details is also crucial if microchipping is to be useful. Lancaster et al. [38] reported that 37% of microchipped dogs entering the RSPCA shelter had problems with the data on the microchip, thus decreasing the chances of successful contact with owners, and subsequent reclaim, by over 20%. This indicates that programs designed to increase the accuracy of microchip data throughout an animal’s life must also be explored.

Regulations to microchip animals prior to sale, and the requirement of including microchip numbers in all advertisements could potentially increase microchipping rates throughout Australia. However, it is advised that the system should allow microchip numbers to be cross-checked by members of the public to deter fraudulent activity, and that programs are designed to ensure information accuracy is maintained through the lifetime of the microchip.

### 4.3. Desexing

The predictors for increasing the likelihood of desexing prior to sale on Gumtree (for dogs over 6 months of age) were; if the dog was microchipped; if the dog was a designer breed or cross breed; being offered for free, or a small cost and; the older the dog the more likely they were to be desexed.

According to AMA, desexing rates in Australia have increased slightly from 77% in 2016 to 81% in 2019 [1,2], far higher than the current study’s Gumtree rate of 26.3% (for dogs over the age of 6 months). Only SA and ACT required desexing by 6 months of age, however, ACT had too few advertisements to be of value, and SA desexing requirements do not apply to dogs born prior to 1 July 2018, which limited relevant advertisements to a single advertisement. The data collected in this study should therefore be used as a benchmark for future studies to identify changes to this rate as more dogs fall under new legislation. Whether this low desexing rate is unique to Gumtree, or characteristic of online platforms is unknown, therefore using this study’s benchmark to compare other online markets is also recommended.

The impact of regulations vs. the power of normative beliefs should be studied further to truly understand the greatest factors in gaining higher rates of desexing in each state and territory. This current study provides baseline data to evaluate the impact of relatively new desexing regulations in SA, and for any states and territories who implement similar policies in the future.

### 4.4. Breeder Identification Numbers

The main predictors for increasing the likelihood of an advertisement including a breeder ID number on Gumtree from SA, QLD and VIC were: if an animal was vaccinated; if they were being sold by someone who identified themselves as a breeder, and; the younger the dog, the more likely they were to have breeder ID’s included, with the highest rate of breeder ID compliance from sellers advertising animals less than 8 weeks of age.

It is important to remember that although all three states included in the modelling have regulations relating to breeder ID numbers, they differ widely. South Australia and QLD requires ID numbers for all dogs born after regulations came into effect while Victoria only requires breeders with more than three fertile dogs to obtain an ID number. Interestingly, only 5.1% of Victorian advertisements included a breeder ID number, which is lower than expected considering current ANKC registration numbers [33] and number of Gumtree advertisements. It is possible that breeders are complying but not placing their number in their advertisements, and further research would be required to make any conclusive statements.

Fraudulent online sales are an increasing issue in Australia and overseas [39,40,41]. Registration numbers are designed to increase the transparency of an animal’s history, including where, and by whom, they were bred and raised. They may also help minimise the risk of online scams. Although outside the scope of this study, some QLD supply numbers were cross checked against the Queensland Dog Breeder Register, which is available publicly and found that not all were valid numbers. Additionally, few of the small number of valid QLD registration numbers checked included contact details for the breeder, making it impossible for potential buyers to confirm the seller and the recipient of the supply number was the same person. SA Dogs and Cats Online system allow any member of the public to search for a breeder by location, name, suburb, authorized association (i.e., Dogs SA), or business name. However, while it provides the first and last name of the individual registered, and allows for breeders/sellers to include contact information, not all include a contact number, which, again, can make it difficult for buyers to confirm the contact number for the individual registered is the same as the contact number in the sellers ad. Notably, Victoria’s new Pet Exchange Registry also does not provide buyers access to contact details for supplier number holders, but does provide the name of the local council the buyer resides in.

The purpose of breeder ID’s is similar to that of microchip numbers—to increase transparency and track the supply of pet dogs in Australia to breeders. However, this study has found that there are some limitations on the usefulness of breeder ID regulations for the average buyer (although it may have a use for other stakeholders). Not being able to confirm the validity of breeder ID’s, and the fact that breeder ID’s are not a guarantee of welfare is an issue, and the inclusion of an ID in an advertisement may lead to false assumptions on the part of the buyer. An Australia-wide system is likely the option that will receive the greatest benefit from a compliance and welfare standpoint. Registry’s should include; species, breed, sex, age, original breeder ID number and contact (number or email), age, desexing status and industry watchdogs should be able to place cautions and history information when breeders have been previously found less than reputable. This would allow buyer to identify less than reputable breeder ID’s at a glance whilst also confirming the identity of the online seller and the dog demographics being sold.

### 4.5. Limitations and Future Research

It is important to recognise that this study relied on the honesty of the seller as well as the assumption that sellers include regulation compliance key words in their advertisement (i.e., microchipped). It is likely that there will be some sellers appearing compliant without doing so, while others may comply but not state it in their advertisements. In addition, the webscraping was only performed two times over a short period of time and may not be representative of Gumtree advertisements posted over a year.

Although state and territories also regulate breeds, this area was outside the scope of this study. Several restricted breeds were identified in the data available for sale, indicating a potential for future research in this area.

Since this study only explored a single online advertising platform, it has some limitations. It is likely that the data obtained from this study is typical for general trading sites similar to Gumtree, such as the Trading Post, but without further research this cannot be confirmed. However, the authors of this study expect that specialist and social sites would differ in their compliance levels. Websites who host predominately sellers identifying themselves as breeders (i.e., Dogzonline.com.au who only allow active, financial members of ANKC association and limited rescue associations to advertise) would be expected to show higher rates of microchipped dogs and breeder ID numbers, while more casual, social platforms, such as Facebook, would likely show lower rates of compliance overall. Further research is recommended.

## 5. Conclusions

This is the first Australian study to our knowledge to benchmark regulation compliance in online (Gumtree) advertising, as well as aiming to provide evidence on factors related to regulatory compliance in online advertisements. Ultimately, this will allow states and territories to make evidence-based decisions for future changes in regulation to encourage higher rate of compliance. The regulations of online advertising in Australia varies widely between states and territories, are inadequate for the size of the market, and is currently insufficient to allow proper animal traceability. The model Victoria has developed may enable better traceability and the benchmarking this study has presented will allow future analysis of the effectiveness of these and future regulation changes.

## Figures and Tables

**Figure 1 animals-10-00425-f001:**
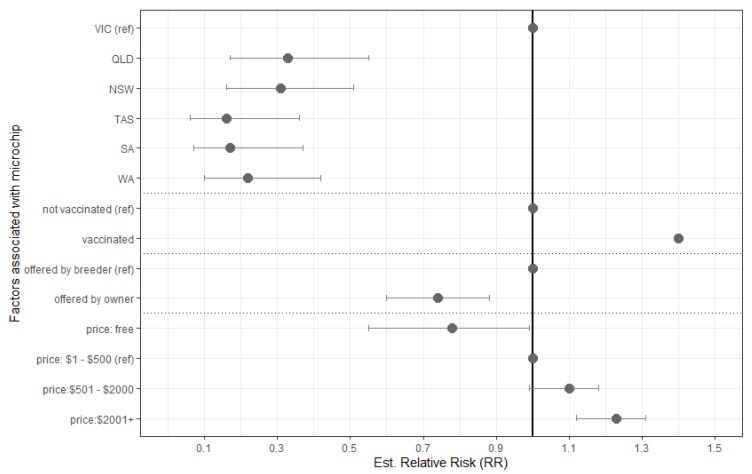
Estimated relative risk for predictors associated with dogs being microchipped at time of sale in 1735 unique online advertisements posted on Gumtree (webscraped on two separate days, 25 March and 8 April 2019), as found through multivariable logistic regression model. Predictors remaining were statistically significant to *p* < 0.005. Desexing was removed from the model due to interactions likely from the differences in procedural age.

**Figure 2 animals-10-00425-f002:**
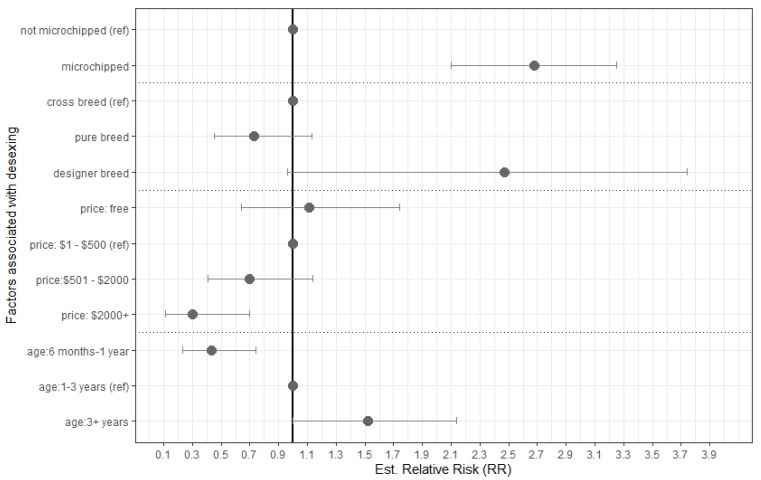
Estimated relative risks for predictors associated with dogs being desexed at time of sale in 499 unique online dog and puppy advertisements posted on Gumtree, webscraped on two separate days (25 March and 8 April 2019) as found through multivariable logistic regression model. Predictors remaining were statistically significant to *p* < 0.005. Dataset only includes dogs over 6 months of age.

**Figure 3 animals-10-00425-f003:**
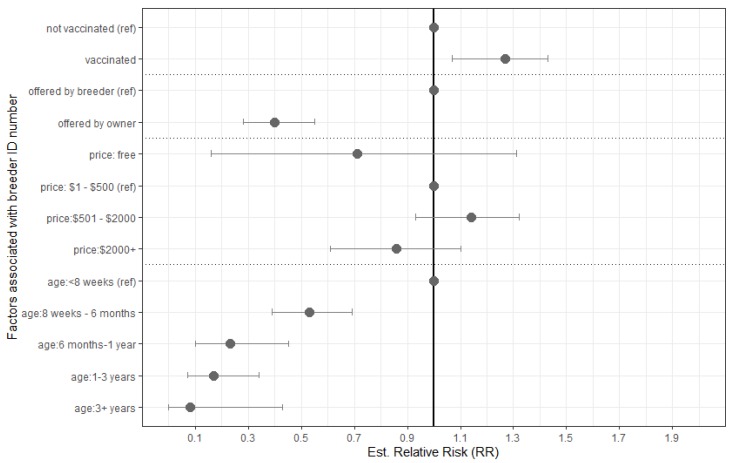
Estimated relative risk for predictors associated with advertisements including a breeder ID at time of sale in 648 unique online advertisements posted on Gumtree (webscraped on two separate days, 25 March and 8 April 2019), in states with regulations relating to breeder ID (SA, QLD, VIC) as found through multivariable logistic regression. Dataset excludes SA advertisements featuring dogs born prior to 1 July 2018, and QLD advertisements featuring dogs born prior to 26 May 2017 as breeder ID regulations did not apply. Predictors remaining were statistically significant to *p* < 0.005.

**Table 1 animals-10-00425-t001:** Regulations for dog and puppy trade (including online) per state and territory in Australia as of 25 March 2019.

State	Desexing	Microchipping	Breeder Registration
Queensland (QLD) *Animal Management (Cats and Dogs) Act 2008*	-	Prior to sale if born after 10 April 2009	Supply number required if dog/puppy born on or after 26 May 2017.
New South Wales (NSW) *Companion Animal Act 1998*	-	Prior to sale	-
Australian Capital Territory (ACT) *Domestic Animals Act 2000*	By 6 months of age	Prior to sale	Breeding licence must appear on all advertisements
Victoria (VIC) *Domestic Animal Act 1994* *Domestic Animals Amendment (Puppy Farms and Pet Shops) Act 2017*	-	Prior to sale. Microchip number must be included in ad	Required if breeder has more than three fertile dogs
Tasmania (TAS) *Dog Control Act 2000*	-	By 6 months of age	-
South Australia (SA) *Dog and Cat Management Act 1995*	By 6 months of age if born after 1 July 2018, or within 28 days of acquisition.	Prior to sale	DACO ^1^ registration or ANKC ^2^ number required on all advertisements
Western Australia (WA) *Dog Act 1976*	-	By 3 months of age	-
Northern Territory (NT) *Nil*	-	-	-

^1^ Dog and Cats Online (breeder registration database); ^2^ Australian National Kennel Council

**Table 2 animals-10-00425-t002:** Frequency per state of data acquired from two individual webscrapes from Gumtree dog and puppy advertisements (25 March and 8 April 2019) in Australia. Total of 1735 unique advertisements. For all rows, N equals ‘number of advertisements’ row, unless otherwise stated.

Variables		QLD	NSW	ACT	VIC	TAS	SA	WA	NT	Unknown	Nationally
General frequencies											
	Number of advertisements	497	575	7	177	58	99	138	20	164	1735
	Number of advertisements per capita (1000) [32]	0.097	0.071	0.016	0.026	0.108	0.056	0.052	0.081	-	0.068
Vet work											
	Microchipped (%)	76.3%	72.0%	42.9%	92.7%	50.0%	68.7%	62.3%	20%	44.5%	70.3%
	Microchip number included in ad	2.8%	1.9%	0.0%	69.5%	5.2%	6.1%	1.4%	0.0%	8.5%	10.0%
	Desexing ^1^	31.8% (N = 154)	26.9% (N = 167)	0.0% (N = 3)	24.2% (N = 62)	27.3% (N = 22)	26.1% (N = 23)	23.3% (N = 43)	50% (N = 2)	13.6% (N = 59)	26.3% (N = 537)
	Vaccination	78.5%	76.7%	71.4%	81.4%	62.1%	77.8%	68.8%	75.0%	54.9%	74.5%
Breed details											
	Cross	33.2%	24.9%	57.1%	20.3%	32.8%	19.2%	37.7%	55.0%	30.5%	28.8%
	Pure	57.1%	67.1%	28.6%	67.8%	63.8%	70.7%	56.5%	25.0%	62.8%	62.5%
	Designer	8.2%	7.7%	14.3%	10.7%	3.4%	6.1%	3.6%	10.0%	3.7%	7.3%
	Unknown	1.4%	0.3%	0.0%	1.1%	0.0%	4.0%	2.2%	10.0%	3.0%	1.4%
Age details											
	<8 weeks	37.5%	35.3%	14.3%	22.0%	34.5%	49.5%	34.8%	30.0%	27.4%	34.5%
	8 weeks–6 months	31.2%	35.7%	42.9%	42.9%	27.6%	27.3%	34.1%	50.0%	36.6%	34.5
	6 months–1year	7.6%	10.3%	28.6%	7.9%	19.0%	6.1%	5.8%	10.0%	11.6%	9.2%
	1–3 years	13.5%	9.7%	0.0%	12.4%	12.1%	6.1%	17.4%	5.0%	15.2%	12.0%
	3–7 years	5.8%	5.2%	0.0%	6.8%	5.2%	4.0%	7.2%	0.0%	4.3%	5.5%
	7+ years	1.6%	0.7%	14.3%	0.6%	0.0%	0.0%	0.7%	0.0%	2.4%	1.1%
	Unknown	2.4%	3.1%	0.0%	7.3%	1.7%	7.1%	0.0%	5.0%	2.4%	3.2%
Price details											
	Free	5.0%	5.6%	14.3%	8.5%	5.2%	3.0%	3.6%	5.0%	8.5%	5.7%
	$1–$500	34.8%	36.7%	42.9%	16.9%	60.3%	29.3%	39.9%	60.0%	39.6%	35.3%
	$501–$2000	36.4%	33%	28.6%	37.9%	12.1%	39.4%	31.9%	30.0%	31.7%	33.9%
	$2001+	21.3%	19.7%	14.3%	31.1%	17.2%	25.3%	21.0%	0.0%	12.2%	20.7%
	Unknown	2.4%	5.0%	0.0%	5.6%	5.2%	3.0%	3.6%	5.0%	7.9%	4.4%
Seller details											
	Breeder	50.7%	37.9%	28..6%	51.4%	22.4%	57.6%	32.6%	20.0%	18.9%	41.1%
	Owner	42.9%	60.9%	71.4%	48.0%	77.6%	40.4%	65.2%	80.0%	80.5%	56.3%
	Shelter/rescue	6.4%	1.2%	0.0%	0.6%	0.0%	2.0%	2.2%	0.0%	0.6%	2.7%
	Registered breeder ID included ^2^	64.6%	8.0%	0.0%	20.9%	6.9%	53.5%	10.1%	0.0%	0.0%	27.4%
	ANKC registered	11.5%	10.4%	0.0%	20.3%	8.6%	17.2%	13.0%	0.0%	1.2%	11.2%
	Other breeder organisation registered	17.5%	20.7%	28.6%	17.5%	5.2%	9.1%	15.2%	0.0%	14.0%	17.0%

^1^ For dogs over six months old; ^2^ either state/territory registration ID (i.e., DACO number) or Australian National Kennel Council member number.

**Table 3 animals-10-00425-t003:** Price and age distribution from two individual webscrapes from Gumtree dog and puppy advertisements (25 March and 8 April 2019) in Australia. Total of 1735 unique advertisements, 76 advertisements did not include a price and 56 advertisements did not include age of dog.

Breed Type	Median Price (AU$)	Price Range (AU$)	<8 Weeks (%)	8 Weeks to 6 Months (%)	6 Months to 1 year (%)	1–3 Years (%)	3–7 Years (%)	7+ Years (%)	Age Unknown (%)
Purebred (N = 1085)	1100	0–10,000	33.9	34.9	8.8	12.6	5.2	0.8	3.8
Cross (N = 499)	395	0–3000	34.3	31.1	11.2	12.0	7.2	1.4	2.8
Designer (N = 126)	2200	0–5000	45.2	45.2	5.6	1.6	1.6	0.0	0.8
Unknown (N = 25)	90	0–22,000	12.0	32	4.0	36.0	4.0	12	0.0
Overall	800	0–22,000	34.5	34.5	9.2	12	5.5	1.1	3.2

**Table 4 animals-10-00425-t004:** Univariate analysis of factors associated with dogs being microchipped prior to sale from two individual webscrapes from Gumtree dog and puppy advertisements (25 March and 8 April 2019) in Australia. Australian Capital Territory and Northern Territory were excluded due to insufficient counts, rescue and shelter advertisements were excluded due to interactions, leaving a total of 1662 unique advertisements for analysis. Predictors have been grouped for ease of interpretation with the reference indicated with a (ref) and a relative risk (RR) of 1. A *p*-value of <0.05 was considered significant.

Variables		Not Microchipped		Microchipped		Total			
		n	%	n	%	n	RR	(95% RR)	*p*-Value
States									
	VIC (ref)	13	7	163	93	176	1.00		
	QLD	115	25	350	75	465	0.52	(0.33–0.72)	<0.001
	NSW	158	28	410	72	568	0.47	(0.29–0.66)	<0.001
	TAS	29	50	29	50	58	0.23	0.11–0.41	<0.001
	SA	30	31	67	69	97	0.42	(0.24–0.65)	<0.001
	WA	52	39	83	61	135	0.33	(0.19–0.52)	<0.001
	Unclear	91	56	72	44	163	0.19	(0.10–0.31)	<0.001
	Total	488	29	1174	71	1662			
Vaccination									
	Yes	134	10.9	1100	89.1	1234	1.40	(1.40–1.40)	<0.001
	No (ref)	354	82.7	74	17.3	428	1.00		
	Total	488	29.4	1174	70.6	1662			
Pure, cross and designer breeds									
	Cross (ref)	194	43.3	254	56.7	448	1.00		
	Pure	270	25.2	800	74.8	1070	1.20	(1.15–1.24)	<0.001
	Designer	11	9.0	111	91.0	122	1.34	(1.29–1.38)	0.001
	Unknown	13	59.1	9	40.9	22	0.79	(0.48–1.06)	0.151
	Total	488	29.4	1174	70.6	1662			
Seller (offered by)									
	Breeder (ref)	75	10.6	632	89.4	707	1.00		
	Owner	413	43.2	542	56.8	955	0.39	(0.31–0.46)	<0.001
	Total	488	29.4	1174	70.6	1662			
Breeder organisation affiliation									
	None (ref)	450	38.2	727	61.8	1177	1.00		
	Yes	38	7.8	447	92.2	485	1.34	(1.31–1.36)	<0.001
	Total	488	29.4	1174	70.6	1662			
Age									
	<8 weeks (ref)	131	22	459	78	590	1.00		
	8 weeks–6 months	140	24	433	76	573	0.96	(0.87–1.04)	0.369
	6 months–1 year	62	42	85	58	147	0.69	(0.55–0.82)	<0.001
	1–3 years	91	47	103	53	194	0.62	(0.50–0.74)	<0.001
	3+ years	49	47	55	53	104	0.62	(0.47–0.77)	<0.001
	Unknown	15	28	39	72	54	0.91	(0.70–1.10))	0.351
	Total	488	29	1174	71	1662			
Price									
	Free	60	63	36	38	96	0.76	(0.60–0.91)	0.00112
	$1–$500 (ref)	247	44	311	56	558	1.00		
	$501–2000	122	21	455	79	577	1.24	(1.20–1.28)	<0.001
	$2001 +	32	9	324	91	356	1.35	(1.32–1.37)	<0.001
	Unknown	27	36	48	64	75	1.09	(0.96–1.20)	0.17649
	Total	488	29	1174	71	1662			

**Table 5 animals-10-00425-t005:** Univariate analysis of factors associated with dogs being desexed prior to sale from two individual webscrapes from Gumtree dog and puppy advertisements (25 March and 8 April 2019) in Australia. States were amalgamated into states which regulated desexing (QLD, VIC and SA), and states which did not regulate desexing. Australian Capital Territory and Northern Territory were excluded due to insufficient counts. Rescue and shelter advertisements were excluded due to interactions with desexing. Predictors have been grouped for ease of interpretation with the reference indicated with a (ref) and a relative risk (RR) of 1. A *p*-value of <0.05 was considered significant.

Variables		Not Desexed		Desexed		Total			
		n	%	n	%	n	RR	(95% CI)	*p*-Value
States requiring desexing									
	not required (ref)	320	77	97	23	417	1.00		
	required	17	74	6	26	23	1.12	(0.47–2.03)	0.755
	State unclear	51	86	8	14	59	0.58	(0.27–1.05)	0.097
	Total	388	78	111	22	499			
Microchipped									
	No (ref)	192	88	25	11.5	217	1.00		
	Yes	196	70	86	30.5	282	2.21	(1.69–2.76)	<0.001
	Total	388	78	111	22.2	499			
Vaccination									
	No (ref)	179	84.4	33	15.6	212	1.00		
	Yes	209	72.8	78	27.2	287	1.65	(1.22–2.15)	0.002
	Total	388	77.8	111	22.2	499			
Pure, cross and designer breeds ^1^									
	Cross (ref)	106	73.6	38	26.4	144	1.00		
	Pure	267	80.7	64	19.3	331	0.72	(0.49–1.05)	0.086
	Designer	7	63.6	4	36.4	11	1.41	(0.46–2.77)	0.476
	Unknown	8	61.5	5	38.5	13	1.50	(0.56–2.76)	0.354
	Total	388	77.8	111	22.2	499			
Seller									
	Breeder (ref)	70	88.6	9	11.4	79	1.00		
	Owner	318	75.7	102	24.3	420	1.87	(1.19–2.75)	0.014
	Total	388	77.8	111	22.2	499			
Breeder organisation affiliation									
	None (ref)	343	76.4	106	23.6	449	1.00		
	Yes	45	90.0	5	10.0	50	0.42	(0.15–0.88)	0.034
	Total	388	77.8	111	22.2	499			
Age									
	6 months–1 year	129	88	18	12	147	0.49	(0.28–0.80)	0.004
	1–3 years (ref)	146	75	48	25	194	1.00		
	3 + years	66	63	38	37	104	1.50	(1.03–2.05)	0.033
	Unknown	47	87	7	13	54	0.52	(0.22–1.01)	0.070
	Total	388	78	111	22	499			
Price									
	Free	58	71	24	29	82	1.07	(0.67–1.59)	0.767
	$1–$500 (ref)	137	72	52	28	189	1.00		
	$501–2000	118	83	25	17	143	0.62	(0.38–0.96)	0.033
	$2001 +	47	89	6	11	53	0.39	(0.15–0.82)	0.018
	Unknown	28	88	4	13	32	0.44	(0.13–1.01)	0.080
	Total	388	78	111	22	499			

^1^ Fisher’s exact test.

**Table 6 animals-10-00425-t006:** Univariate analysis of factors associated with advertisements listing breeder ID’s from two individual webscrapes from Gumtree dog and puppy advertisements (25 March and 8 April 2019) in Australia. Only states with breeder ID regulations were included in this analysis (QLD, SA, VIC). SA advertisements only included those selling animals born prior to 1 July 2019, QLD advertisements only include animals born on or prior to 26 May 2017, as ID regulations do not apply to older animals. Dogs with no age details and shelter/rescue advertisements have been excluded from all states. Predictors have been grouped for ease of interpretation with the reference indicated with a (ref) and relative risk (RR) of 1. A *p*-value of <0.05 was considered significant.

Variables		No Breeder ID Included		Breeder ID Included		Total			
		n	%	n	%	n	RR	(95% RR)	*p*-Value
Microchipped									
	No (ref)	81	67	40	33	121	1.00		
	Yes	194	37	333	63	527	1.43	(1.32–1.53)	<0.001
	Total	275	42	373	58	648			
Vaccination									
	No (ref)	83	68.6	38	31.4	121	1.00		
	Yes	192	36.4	335	63.6	527	1.46	(1.35–1.54)	<0.001
	Total	275	42.4	373	57.6	648			
Pure, cross, and designer breeds ^1^									
	Cross (ref)	80	49.4	82	50.6	162	1.00		
	Pure	166	40.5	244	59.5	410	1.15	(1.00–1.28)	0.053
	Designer	19	29.2	46	70.8	65	1.32	(1.11–1.49)	0.006
	Unknown	10	90.9	1	9.1	11	0.20	(0.01–0.72)	0.028
	Total	275	42.4	373	57.6	648			
Seller									
	Breeder (ref)	93	24.3	289	75.7	382	1.00		
	Owner	182	68.4	84	31.6	266	0.29	(0.22–0.38)	<0.001
	Total	275	42.4	373	57.6	648			
Breeder organisation affiliation									
	None (ref)	217	51.5	204	48.5	421	1.00		
	Yes	58	25.6	169	74.4	227	1.40	(1.30–1.49)	<0.001
	Total	275	42.4	373	57.6	648			
Age									
	<8 weeks (ref)	61	22	213	78	274	1.00		
	8 weeks–6 months	109	43	143	57	252	0.59	(0.45–0.74)	<0.001
	6 months–1 year	37	82	8	18	45	0.13	(0.06–0.27)	<0.001
	1–3 years	56	88	8	13	64	0.09	(0.04–0.18)	<0.001
	3+	12	92	1	8	13	0.05	(0.00–0.25)	<0.001
	Total	275	42	373	58	648			
Price									
	Free	23	92	2	8	25	0.23	(0.04–0.61)	0.003
	$1–$500 (ref)	93	56	72	44	165	1.00		
	$501–$2000	82	31	180	69	262	1.38	(1.25–1.48)	<0.001
	$2001 +	64	36	113	64	177	1.31	(1.16–1.44)	<0.001
	Unknown	13	68	6	32	19	0.78	(0.37–1.19)	0.317
	Total	275	42	373	58	648			

^1^ Fisher’s exact test.

**Table 7 animals-10-00425-t007:** Benchmark of total, and regulation compliance, in microchipping, desexing and breeder registration in Gumtree dog and puppy advertisements from two individual webscrapes (25 March and 8 April 2019) per state in Australia. Refer to Table 1 for regulations for each state and territory.

Variables	QLD	NSW	ACT	VIC	TAS	SA	WA	NT	State Unknown	Total
**Microchip—total**	76.3% (N = 497)	72% (N = 575)	42.9% (N = 7)	92.7% (N = 177)	50.0% (N = 58)	68.7% (N =99)	62.3% (N = 138)	20% (N = 20)	44.5% (N = 164)	70.3% (N = 1735)
**Microchip compliance**	76.8% ^1^ (N = 494)	72% ^2^ (N = 575)	42.9% (N = 7)	92.7% (N = 177)	59.1% ^3^ (N = 22)	68.7% (N = 99)	52.5% ^4^ (N = 59)	-	-	-
**Microchip number included in ad—total** ^5^	2.8% (N = 497)	1.9% (N = 575)	0.0% (N = 7)	69.5% (N = 177)	5.2% (N = 58)	6.1% (N =99)	1.4% (N = 138)	0.0% (N = 20)	8.5% (N = 164)	10.0% (N = 1735)
**Desexed—total** ^6^	31.8% (N = 154))	26.9% (N = 167)	0% (N = 3)	24.2% (N = 62)	27.3% (N = 22)	26.1% (N =23)	23.3% (N = 43)	50% (N = 4)	13.6% (N = 59)	26.3% (N = 537)
**Desex compliant**	-	-	0% ^7^(N=3)	-	-	100% ^8^ (N=1)	-	-	-	-
**State breeder ID included—total** ^9^	59.6% (N = 497)	-	0.0% (N = 7)	5.1% (N = 177)	-	53.5% (N =99)	-	-	-	
**Breeder ID compliant**	69.9% ^10^ (N = 406)	-	0.00% (N = 7)	* ^11^	-	64.9% ^12^ (N = 77)	-	-	-	-

^1^ Exemptions: dogs born prior to 10 April 2009; ^2^ although working dogs kept in the Western Division of NSW that do not fall within local government area or on land rated as farmland are exempted, data was insufficient to confirm, therefore all working dogs are included in this case; ^3^ required by 6 months of age, exemptions: working livestock dogs, registered hunting dogs and racing greyhounds; ^4^ required by 3 months of age; ^5^ only VIC requires all advertisements to include microchip numbers; ^6^ advertisements for dogs over the age of six months. ^7^ required by 6 months of age, exemptions: dogs born prior to 21 June 2001; ^8^ required by 6 months of age, exemptions: dogs born prior to 1 July 2018, working livestock dogs, racing greyhounds and animals deemed a high surgical risk by a licensed veterinarian; ^9^ state breeder registration ID only (i.e., DACO number), ANKC numbers not included; ^10^ valid breeder ID includes BIN number, ANKC number, Logan City Council number, Gold Coast City Council number, Queensland Racing Integrity Commission number and Breeder Exemption number, exemptions: dogs born prior to 26 May 2017 and working livestock dogs; ^11^ data collected insufficient to confirm compliance of breeders with three or more fertile dogs. Breeders not required to include ID’s in advertisements; ^12^ valid breeder ID includes Dogs and Cats Online registration numbers and ANKC member numbers, exemptions: dogs born prior to 1 July 2018.

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
