# Peer review of "Regulatory Compliance in Online Dog Advertisements in Australia"

_animals, 2020, doi:10.3390/ani10030425_

Round 1
Reviewer 1 Report
This is a very interesting manuscript on an important pet welfare topic with limited regulation and less research or enforcement.
Summary/Abstract: please add the term “general” or something similar to the descriptor of Gumtree to make clear this isn’t animal specific at all. That isn’t clear unless you look it up or read the full manuscript.
Reference 1: the link needs to be updated. And the 2019 Australian pet ownership data should be used.
Reference 10: the reference has a typo “Availabe”
Table 1 should be closer to where it is first cited in the text.
Line 77: need a semi-colon after [20].
Paragraph starting line 119: can the authors please include some info about how many where removed for these reasons in the results? That gives some context to understand the potential bias from missing or incorrect data.
Please make the order of the locations variables the same in Tables 1 and 2.
Section 2.3: please include how/why some interactions were examined and determined to be included or not. And did logistic regression models meet the assumptions? Please list assumptions/fit assessment here and confirm in the results. Why was dog breed not included using the most common ones and then lumping the rest into other? Size of dog was also mentioned but didn’t seem to be included anywhere in the manuscript analysis?
Line 133: Yates correction for continuity is rarely used these days and mostly for data sets when the expected value is small or there is a small sample, probably not the case for these analyses. Please reference a recent recommendation or change. A univariable logistic regression would also be fine here.
Line 142-3: which variables had this done to them? Again, this information influences context and interpretation.
Table 2: the discussion provides some context for the number of breeders. I think that adding a row here to show the number of advertisements per 1000 people or households would provide valuable context as well. I also would like to see a comment in the discussion about selling puppies < 8 weeks old. Granted they may not go to their new homes until after 8 weeks but please clarify if that was the situation or if some of these puppies likely were sold to new homes before 8 weeks of age and the potential problems with doing that. I would anticipate that less knowledgeable or scrupulous sellers are placing puppies in homes too young.
Line 184-5: please number tables in the order in which they are listed in the text.
For tables 4-6, please put the locations (even if categorized) at the top of the list of variables so that the variable sequence in that column is consistent.
Line 227: I think that the phrase “…reference [1]” contains an error.
Line 228 and following: why was the interaction not included in the model? Because it created zero cells and the model didn’t converge? Otherwise, I don’t see why it isn’t worth including for a more accurate picture.
Line 258-9: Does this mean the interaction was examined for statistical significance and was not significant? I’m a bit confused by how this was phrased if it was explored statistically or for real world relevance of the interaction.
331-2: Do the authors believe that non-compliance is due to difficulties getting to the vet? Seems like the puppy would go for vaccinations and get chipped at that time. I would guess either that cost of the chip or the vet not bringing it up are more important. Please discuss in the manuscript.
Line 337 and 342: if the microchipping rate is 87%, how is the stray dog microchipping rate only 28%? Please discuss that it seems likely that dogs who become stray are in households less willing, able or knowledgeable about microchipping or include whatever explanation the authors believe to explain this substantial difference.
Section 5.4: any limitations due to missing or impossible data being interpolated or deleted? Please include. Also, any big differences likely to be found between Gumtree and other online sales options in the authors’ opinions? That expert opinion is helpful.
Author Response
Reference 1: the link needs to be updated. And the 2019 Australian pet ownership data should be used.
Have updated the first part of this sentence so it reflects and references the 2019 report. The second part of this sentence I have kept the 2016 report as this information does not appear in the 2019 version (only the general ‘pet ownership’, not dog ownership specifically). References have been fully updated to reflect these changes.
Also updated industry size to reflect data from 2019 report. Have updated all other references to the 2016 report to the 2019 report similarly excluding the below sentences:
“The results in the current study reflect AMA’s 2016 report where they reported Victoria had the highest rate of microchipping at 89% [2]” - as again, this specific info does not appear in the 2019 version.
“According to AMA, desexing rates in Australia have increased slightly from 77% in 2016 to 81% in 2019 [1,2]…” – as this compares the changes between both reports.
Reference 10: the reference has a typo “Availabe”
Typo has been amended.
Table 1 should be closer to where it is first cited in the text.
Table 1 has been moved directly below first citation in introduction.
Line 77: need a semi-colon after [20].
Semi-colon has been added.
Paragraph starting line 119: can the authors please include some info about how many where removed for these reasons in the results? That gives some context to understand the potential bias from missing or incorrect data.
Now line 171: “Prices were kept as in the original ad, excluding those obviously artificial, e.g. $1234 (4 records), and those priced at $1 (11 records), which were labelled as missing, leading to total of 52 advertisements with missing price details.”
Please make the order of the locations variables the same in Tables 1 and 2.
Locations have been updated so they are in the same order.
Section 2.3: please include how/why some interactions were examined and determined to be included or not. And did logistic regression models meet the assumptions? Please list assumptions/fit assessment here and confirm in the results. Why was dog breed not included using the most common ones and then lumping the rest into other? Size of dog was also mentioned but didn’t seem to be included anywhere in the manuscript analysis?
Extra information was added to section 2.3 : “Variables with chi-squared test p-values less than 0.25 were included in the regression models. Based on literature and plausibility, the interaction between desexing status and microchipping, due to the age differences of when these procedures typically occur, lead to the decision to not include desexing in the microchipping model.” and “To assess the fit of the multivariable regression models used Akaike Information Criterion (AIC) statistics.”
We have revised the results of the interactions terms to reflect our modelling approach, which was not correct. (Lines 235 and 258 from original version reviewed).
Regarding why dog breed wasn’t included, even with breeds grouped together we found that we didn’t have enough power in the numbers. The authors also were more interested in the pure/cross/designer dynamics than actual differences between breeds.
Size of dog was mentioned as it was considered a unique breed identifier – i.e. we counted a miniature poodle as a different breed than a standard poodle, and different to a toy poodle – leading to counting three breeds represented in the data, rather than one ‘poodle’ breed. We did not do analysis based by size in any other way, i.e. small dogs vs. big dogs. This was outside the scope of this study.
Line 133: Yates correction for continuity is rarely used these days and mostly for data sets when the expected value is small or there is a small sample, probably not the case for these analyses. Please reference a recent recommendation or change. A univariable logistic regression would also be fine here.
Reviewer is correct and wording in this section has been changed to chi-squared and Fisher’s exact tests, which is what was used.
Now line 183 - “Frequency analysis was completed prior to full statistical analysis and chi-squared tests or Fisher’s exact tests were performed on categorical data to test for association between variables.”
Line 142-3: which variables had this done to them? Again, this information influences context and interpretation.
Now line 200: we only made these changes to three variables, have amended the sentence to be clearer. “ As counts were too small for analysis, ‘unknown’ was changed to ‘no’ for microchipping, vaccination and breeder org affiliation.”
Table 2: the discussion provides some context for the number of breeders. I think that adding a row here to show the number of advertisements per 1000 people or households would provide valuable context as well. I also would like to see a comment in the discussion about selling puppies < 8 weeks old. Granted they may not go to their new homes until after 8 weeks but please clarify if that was the situation or if some of these puppies likely were sold to new homes before 8 weeks of age and the potential problems with doing that. I would anticipate that less knowledgeable or scrupulous sellers are placing puppies in homes too young.
An extra row has been added to Table 2, with details on the number of advertisements per capita in each state. References have been updated.
Extra sentences were added to include discussion on selling puppies <8 weeks of age. References were updated accordingly.
Line 380 - “Overall, 34.5% of sellers advertised puppies less than 8 weeks of age, although it was outside the scope of this study to quantify this data, most advertisements for underage puppies specified they would be “available in x weeks”. However, not all had this specification and it is likely that some would have been available prior to reaching 8 weeks of age. It is well documented that early separation from mum risks long-term behavioural concerns [37] and further research on how many underage puppies are available on Gumtree, and other online platforms, is recommended."
Line 184-5: please number tables in the order in which they are listed in the text.
We have amended the sentence so that the tables appear in the same order as they are cited in the text.
Line 242 - “Univariate analysis found that advertisers who identified themselves as breeders were 61% more likely to sell microchipped dogs than those who identified as owners, but owners were 87% more likely to advertise desexed animals compared to those identifying as breeders, see Table 4 and 5. Those who identified themselves as breeders were 71% more likely to include a breeder ID in their advertisement than other types of sellers, see Table 6. “
For tables 4-6, please put the locations (even if categorized) at the top of the list of variables so that the variable sequence in that column is consistent.
Have updated Table 5 so that the (categorised) states are at the top.
Line 227: I think that the phrase “…reference [1]” contains an error.
error has been amended.
Line 228 and following: why was the interaction not included in the model? Because it created zero cells and the model didn’t converge? Otherwise, I don’t see why it isn’t worth including for a more accurate picture.
Have re-written sentence to more accurately and clearly describe the decision-making process:
Line 288 - “Univariate analyses found that desexed dogs were 15% more likely to be microchipped. However, while desexing was found to be a significant factor in the regression model, desexing was subsequently removed from the model due to being highly correlated with the age of the dog.”
Line 258-9: Does this mean the interaction was examined for statistical significance and was not significant? I’m a bit confused by how this was phrased if it was explored statistically or for real world relevance of the interaction.
Apologies for the lack of clarity, we looked at the statistical significance and it was not significant. Authors have removed this sentence considering it was not significant.
331-2: Do the authors believe that non-compliance is due to difficulties getting to the vet? Seems like the puppy would go for vaccinations and get chipped at that time. I would guess either that cost of the chip or the vet not bringing it up are more important. Please discuss in the manuscript.
New line: 396. Have reworded this paragraph to add in discussion on authors thoughts on non-compliance. Please note that due to editors comments, some information found in this section originally, has been moved to the introduction (line 90).
“This study has revealed that dogs advertised as vaccinated are more likely to be microchipped suggesting that sellers who take their animals to the vet are likely to carry out both these procedures, supporting McGreevy, et al. [13] survey findings. Rohlf, et al. [12] thoughts on normative beliefs are also supported by this study’s findings on Gumtree advertisements; Sellers who comply, or pursue “good guardianship” practices, are more likely to comply fully, while those that are either unwilling, or unable to, do not comply at all. Whether non-compliance in Gumtree advertisements is due to financial, unwillingness, or access concerns is unknown and more research would be required before reaching any conclusions.”
Line 337 and 342: if the microchipping rate is 87%, how is the stray dog microchipping rate only 28%? Please discuss that it seems likely that dogs who become stray are in households less willing, able or knowledgeable about microchipping or include whatever explanation the authors believe to explain this substantial difference.
New line 413. Paragraph has been re written.
“In 2013, Lancaster, et al. [38] found only 28% of stray dogs were microchipped when they entered the RSPCA Queensland shelter. This may show that stray dogs are more likely to come from households who are less likely to microchip their dogs, either due to lack of knowledge, financial ability or willingness to do so. It must also be noted that some dogs who are microchipped may bypass the RSPCA, as their microchips can be scanned at local vets (where stray animals are often taken to by members of the public), leaving animals who are more difficult to identify (un-microchipped) more likely to enter the RSPCA system, potentially showing a skewed population. However, correct and updated microchip details is also crucial if microchipping is to be useful. Lancaster, et al. [38] reported that 37% of microchipped dogs entering the RSPCA shelter had problems with the data on the microchip, thus decreasing the chances of successful contact with owners, and subsequent reclaim, by over 20%. This indicates that programs designed to increase the accuracy of microchip data throughout an animal’s life must also be explored.”
Section 5.4: any limitations due to missing or impossible data being interpolated or deleted? Please include. Also, any big differences likely to be found between Gumtree and other online sales options in the authors’ opinions? That expert opinion is helpful.
Paragraph has been added to limitations section to include the discussion requested:
“Since this study only explored a single online advertising platform, it has some limitations. It is likely that the data obtained from this study is typical for general trading sites similar to Gumtree, such as the Trading Post, but without further research this cannot be confirmed. However, the authors of this study expect that specialist and social sites would differ in their compliance levels. Websites who host predominately sellers identifying themselves as breeders (i.e. Dogzonline.com.au who only allow active, financial members of ANKC association and limited rescue associations to advertise) would be expected to show higher rates of microchipped dogs and breeder ID numbers, while more casual, social platforms, such as Facebook, would likely show lower rates of compliance overall. Further research is recommended.”
Reviewer 2 Report
The Authors have conducted an admirable study of the dynamic factors involved, and have addressed its limitations, namely, the common concern being fraudulent online sales in Australia and elsewhere, and the fact that their study relied upon the honesty of the sellers.
The Authors point out that "More than half of all Australian homes are shared with a companion animal, 38% of which do so with at least one dog, compared to 27% of the United Kingdom, and 50% of the United States of America households. With the total population of dogs reaching 5 million nationwide, and increasing by 3% each year, Australia’s love for dogs feeds a 7 billion-dollar industry and requires a supply of approximately 480,000 animals each year to meet demand."
As the Authors state "This is the first Australian study to our knowledge to benchmark regulation compliance in online advertising, as well as aiming to provide evidence on factors related to regulatory compliance in online advertisements. This will allow States and Territories to make evidence-based decisions for future changes in regulation to encourage higher rate of compliance."
Further: "Registry’s should include; species, breed, sex, age, original breeder ID and contact (number or email), age, desexing, status and industry watchdogs should be able to place cautions and history information when breeders have been previously found less than reputable."
Author Response
There were no comments. Thank you for your review.
Reviewer 3 Report
In this study, dog trade advertisements collected from animal trading websites (Gumtree) on two separate dates were analyzed by chi-square test and multivariable logistic regression modeling to determine the following risk factors: microchip, sterilization, breeder identification code, The level of compliance in each of these cases. Because different Australian states/Territories have different legislative standards, some projects are not included in the analysis because some states do not have such a requirement. This is why the results of this study are more complicated. The data contains a total of 1735 valid samples.
Some questions below:
Line 152: and the German Shepherd (5.07 % forget percentage ?).
Line 160: …all states had similar rates of desexing (checked by outlier detection?).
Table 1 lists 8 states/Territories, but Table 7 lists 9 states/Territories. Should Table 1 be added?
Line 225~230: (1)Punctuation is confusing, (2)"-" is not a conjunction, please replace them, using formal statements.
This sentence「advertisements posted on Gumtree, webscraped on two separate days (March 25th and April 8th, 2019)」has been always repeated in the article. Can you define it clearly in advance in materials and methods? Otherwise readers will be tired after repeated reading.
Line 244 March 25th and April 8th, 2019),) Forget punctuation?
Author Response
Line 152: and the German Shepherd (5.07 % forget percentage ?).
Typo fixed. Percentage sign has been added.
Line 160: …all states had similar rates of desexing (checked by outlier detection?).
Apart from the two states with limited number of dogs (ACT and NT) in other states the desexing rates varied between 23.3 (WA) and 31.8 (QLD). Dogs could either be desexed or not (yes/no) and therefore outlier detection was not used.
Table 1 lists 8 states/Territories, but Table 7 lists 9 states/Territories. Should Table 1 be added?
NA was for ‘state unknown’, as there were ads where the state of origin could not be identified. Have clarified Table 7 and changed ‘NA’ to ‘state unknown’ for clarity.
Line 225~230: (1)Punctuation is confusing, (2)"-" is not a conjunction, please replace them, using formal statements.
Have rewritten sentences. Line 288:
“Univariate analyses found that desexed dogs were 15% more likely to be microchipped. However, while desexing was found to be a significant factor in the regression model, desexing was subsequently removed from the model due to being highly correlated with the age of the dog.”
This sentence「advertisements posted on Gumtree, webscraped on two separate days (March 25th and April 8th, 2019)」has been always repeated in the article. Can you define it clearly in advance in materials and methods? Otherwise readers will be tired after repeated reading.
Materials and methods beginning at line 138 defines webscraping procedure.
Presuming this is referring to the repeating of this procedure in each table and figure. Authors believe tables and figures should stand alone with full details of what is contained in them. However, if editor would like this removed, we are happy to comply. Please advise.
Line 244 March 25th and April 8th, 2019),) Forget punctuation?
Typo error fixed.